# OpenReview forum: "[Re] Cooperate or Collapse: Emergence of Sustainable Cooperation in a Society of LLM Agents"
_TMLR — Accepted by TMLR_

### Review · Reviewer_evNU · 2025-04-07

**Summary Of Contributions:**

The authors replicate the results of “Cooperate or Collapse: Emergence of Sustainable Cooperation in a Society of LLM Agents” with small LLMs. They extend the work by also evaluating recently released “reasoning models” and a suite of variations related to universalization.

**Audience:**

No

**Claims And Evidence:**

Yes

**Requested Changes:**

I believe the following new experiments are critical to support the author's claims and have this work stand on its own:

1. Given the interest in reasoning models, I would like to see the authors evaluate the environment with at least one SOTA reasoning model such as R1, o1, claude-thinking etc.

2. The universalization experiments should be replicated with a SOTA model (e.g., GPT-4o). It is unclear if some of the manipulations around Universalization didn’t work because the models are too weak, and thus it is hard to draw a generalizable conclusion. This should be possible on an academic budget.

I believe that if these issues are addressed, the paper will be of interest to the TMLR audience.

**Strengths And Weaknesses:**

Strengths:
The paper is straightforward and transparent, as this is primarily a replication. It doesn’t overclaim and doesn’t try to oversell the results. The results are interesting, and the new experiments are well-designed and discussed appropriately.

Weaknesses:
There is very little new content beyond the replication, as the GovSim environment can be run directly from the authors' open-source release. Thus, replication is less of an example of reimplementing an algorithm from the paper and more of an exercise in running an existing code base.

The original contributions are very incremental.

---

> ### Author Response · Authors · 2025-06-14
> **Response to reviewer evNU**
>
> We appreciate the reviewer’s thoughtful engagement with our work and the helpful suggestions provided.  Please find our responses to the specific points below.
>
> # Weaknesses
> ## W1: Novelty in replication
> Our study offers a more nuanced perspective on the claims made in the original paper. While the authors claim that strong cooperative performance requires large models and that universalization-based reasoning improves GovSim performance, we find that these conclusions require scrutiny. The novelty of our work lies in the systematic exploration of the original implementation, interpretations, and results. Importantly, our findings lead to different conclusions. Specifically, we demonstrate that: (1) Performance correlates more reliably with reasoning-focused training than with model size alone. (2) The sustainability threshold used in the original study complicates and partially obscures the core idea behind universalization. (3) As an alternative example, veil of Ignorance reasoning can enhance performance in larger models, but tends to be less effective, and occasionally even counterproductive, for smaller models.
>
> # Requested Changes:
> We extended our original approach to include DeepSeek-R1 and DeepSeek-V3 as a SOTA models, you can refer to our results section for a full analysis of the new addition. In all, we achieve best-in-class performance on GovSim with R1, outperforming all models from the original paper. The inclusion of these two models in particular is interesting due to their shared architecture, with R1 differing only with its reasoning-focused post-training, allowing us to measure the impact that the reasoning paradigm has on performance.
>
> We also ran both models on all experiments, including the testing of universalization and veil of ignorance. In contrast to the results for the smaller LLMs, we find that these new models are able to reason effectively when prompted with these social reasoning frameworks.
>
> We sincerely thank the reviewer for their comments and critiques, and we are open to further discussion to clarify any aspects of our submission.

---

### Review · Reviewer_xurT · 2025-04-17

**Summary Of Contributions:**

This paper addresses the important issue of cooperation among AI agents in competitive multi-agent environments. The authors propose a method for improving cooperative behavior and provide experiments in several standard benchmarks. The paper is clearly written and the motivation is presented in a meaningful and accessible manner.

**Audience:**

Yes

**Claims And Evidence:**

Yes

**Requested Changes:**

Not quite ready, switch to a different venue for submission.

**Strengths And Weaknesses:**

Strengths:

+ The research question explored is highly relevant, addressing cooperation challenges in contemporary multi-agent and AI safety contexts.

+ The paper is well-written and easy to follow, with clear motivation and methodological exposition.

Weaknesses:

- The methodological approach proposed does not introduce fundamentally novel concepts, relying largely on established techniques with incremental tweaks rather than innovative advances.

- The experiments focus exclusively on open-source models and do not consider prominent closed-source models or more diverse settings. This limits the generalizability and significance of the conclusions, particularly when claims are made about the broader landscape of AI cooperation.

- While the experiments demonstrate some improvements, the results are marginal and not sufficiently compelling to justify the effectiveness of the proposed approach. Stronger baselines or broader benchmarks could have helped support the claims.

---

> ### Author Response · Authors · 2025-06-14
> **Response to reviewer xurT**
>
> We are grateful for the reviewer’s detailed and constructive remarks. We address each comment in detail below.
>
> # Weaknesses
> ## W1: Novelty in methodological approach
> In our work we provide additional nuance to the claims of the original paper. In particular, we find that the two conclusions they make, that larger models are required for good cooperative performance, and that universalization-based reasoning improves GovSim performance, require qualification and can be made more precise. The novelty lies in our systematic analysis of the original implementation, claims, and results. Fundamentally, the takeaways of our two papers are different. We find that:
> 1. Reasoning-focused training is a consistent predictor of performance relative to size
> 2. The original paper's inclusion of the sustainability threshold obfuscates the true concept of universalization
> 3. Other social reasoning methods are effective for larger models, but less-so for smaller ones, and that for these smaller models, their inclusion could even be detrimental (as in the case of Veil of Ignorance)
>
> ## W2:  Focus on open-source models
> While we did not test closed-source models, we evaluated a wide range of small and mid-sized models, and since our original submission, we have added additional benchmarks for DeepSeek-R1 and DeepSeek-V3. This broadens our discussion to larger and state-of-the-art models, which has provides us to see additional trends in the larger models being able to reason effectively with the Veil of Ignorance experiment. Furthermore, the inclusion of these also allowed us to concretely verify that LLM scale is important for GovSim performance, as the original authors claim, but additionally the importance of the recent reasoning paradigm in cooperative decision making.
>
> ## W3: Marginal results
> Regarding the marginal improvements demonstrated in our experiments, our primary goal was to add nuance to the claim that smaller models were unable to perform well on the task, as well as investigate the causes of Universalization to improve model performance: whether the mere presence of the threshold of collapse value led to this improvement, or if the models truly became more cooperative because of the message. We think that with the addition of the SOTA models, this added nuance to the original paper could be of value to any reader in the field of multi-agent cooperation and AI safety. We believe these comparisons contribute to a deeper understanding of model cognition in the field of sustainable cooperation.
>
> We acknowledge and appreciate the reviewer’s time and effort in evaluating our submission, we are open to further discussion to clarify any aspects of our submission.

---

### Review · Reviewer_eRPT · 2025-06-02

**Summary Of Contributions:**

The paper presents a comprehensive replication and extension of a previous work (“Piatti et al., “Cooperate or Collapse: Emergence of Sustainable Cooperation in a Society of LLM Agents”, NeurIPS 2024) which studies the emergence of sustainable cooperation in multi-agent environments modeled with Large Language Models. The authors successfully reproduce the original findings that small, general-purpose open-weight LLMs (e.g., Llama2, Llama3, Mistral) perform poorly in common-pool resource scenarios when acting as autonomous agents.

In addition, the paper contributes two main novel experimental extensions: the evaluation of more models and the exploration of alternative prompting strategies.

In particular, they test some reasoning-enhanced models (Phi4, Qwen2.5-Math-7B, and DeepSeek-R1-Distill-Llama-8B) and demonstrate that smaller models trained with specific attention to reasoning abilities can outperform larger general-purpose models in sustaining cooperation. This insight refocuses the previous findings from the assumption that smaller model size equates to worse performance, highlighting instead that the model's training focus plays a far more significant role in determining cooperative effectiveness.

Regarding the alternative prompting strategies, the authors evaluate two variations of the original universalization prompt: one supplying a sustainability threshold without moral framing, and another encouraging models to assume that others will act like themselves, omitting threshold values. They also examine the impact of decision-making without self-identity, inspired by the Veil of Ignorance theory. The findings indicate that the improvements previously attributed to universalization reasoning may partly result from explicit access to threshold values, while the Veil of Ignorance prompt analysis yields performance gains only for certain models, especially DeepSeek.

Overall, the authors provide a robust reproduction of the previous work adding novel and significant elements of analysis, specifically regarding the role of LLMs’ reasoning abilities in multi-agent cooperation scenarios.
Notably, they also provide some information regarding the difficulties encountered during the reproduction of the original paper, which is extremely relevant to ensure actual reproducibility in academic outputs.

**Audience:**

Yes

**Broader Impact Concerns:**

While the Discussion section may benefit from the addition of a broader impact statement addressing the work’s relevance with respect to real-world contexts, inserting a dedicated section (as “Broader Impact Statement”) is not necessary.

**Claims And Evidence:**

Yes

**Requested Changes:**

A short part in the Discussion section highlighting the broader social relevance of multi-agent LLM cooperation, and the specific contribution of this work in that context, would help demonstrate its impact and importance. This addition could be particularly valuable for readers whose expertise lies outside of game theory and social sciences, offering a clear understanding of why reproducing and extending prior work in this area is important.

No other specific changes are requested for this work.

**Strengths And Weaknesses:**

Strengths
The paper deals with the relevant topic of aligning LLMs for cooperative behavior, which is crucial for multi-agent AI systems and their deployment in real scenarios. The authors do so by faithfully replicating the original significant work by Piatti et al. and by increasing the breadth of the analysis.
- The reproduction work is strong, as it uses the same models and platform as the original one, providing additional validation of prior claims.
- The extension part is also robust and relevant as it introduces newer models, specifically trained for reasoning tasks, and tests alternative prompting strategies to validate previous results and to link the work with broader philosophical theories, increasing its potential impact.
- The presentation is clear and clean. The reproduction process is rightly detailed, and the additions are duly motivated and framed.

Weaknesses
The main weaknesses are related to the limitations already properly addressed by the authors in the dedicated section of the paper.
They concern the narrow focus on small models (due to computational constraints), the prompt engineering sensitivity of LLMs, and the mixed nature of the Veil of Ignorance results.
While meaningful, these weaknesses do not impact the value and validity of the presented work.

---

> ### Author Response · Authors · 2025-06-14
> **Response to reviewer eRPT**
>
> We thank the reviewer for their insightful comments and constructive feedback. Below, we address the
> questions and concerns raised by the reviewer.
>
> # Weaknesses
> ## W1: Narrow focus of small models
> As the reviewer noted, our original results were limited to models that we could benchmark given limited computational resources. However since our original submission, we have been able to run additional experiments using the larger SOTA models: DeepSeek-V3 and DeepSeek-R1. These additions are particularly interesting, since they utilize the same architecture, and only differ by additional post-training in the case of R1. This allowed us to both evaluate the impact of scale on GovSim, as well as evaluate the new reasoning paradigm in LLMs, akin to an ablation study.
> ## W2: Prompt engineering sensitivity of LLMs
> The sensitivity of LLMs to the prompt is a well-known issue. In our case we primarily wanted to show that the type of prompt given can influence how a model behaves, testing different social reasoning methods. We attempted to control for sensitivity by gradually increasing the level of abstraction from the original universalization experiment, which included the sustainability threshold, down to the most abstract 'systemic' prompt.
> ## W3: Mixed nature of the Veil of Ignorance results.
> Our original results showed that Veil of Ignorance was too difficult a concept for the smaller LLMs to grasp. However, with our new inclusions, the larger LLMs show improved performance across all experiments, including Veil of Ignorance, which suggests that the newest SOTA LLMs can benefit from different social reasoning methods.
>
> # Requested Changes:
> To emphasize the societal impact of our results, we've added a section in the discussion section where we discuss multiple real-life examples of multi-agent AI systems, placing our results in context. The additional discussion can be found in section 5.3 on page 11.
>
> We believe that with our additional experiments, and the additional context given by our discussion of real-world examples of multi-agent systems, we provide more nuanced insights into multi-agent reasoning. We sincerely appreciate the reviewer’s time and effort in evaluating our submission and welcome any further questions or comments to help clarify aspects that may require additional discussion.

---

### Decision · Action_Editor_3JfS · 2025-08-18

**Recommendation:** Accept as is

**Audience:**

Yes

**Audience Explanation:**

This paper reproduces and extends prior work, providing deeper insights into earlier studies.

**Claims And Evidence:**

Yes

**Claims Explanation:**

This study examines cooperation strategies among LLM agents in common pool resource environments, replicating prior results on Cooperate or Collapse (Piatti et al., 2024). The authors extend the analysis to newer reasoning-oriented models (Phi-4, DeepSeek-R1, and a distilled variant), and find that these specialized models outperform general-purpose ones of similar scale but require more computation. The study also tests mechanisms such as the veil of ignorance and universalization-based prompting, finding that older models rely heavily on explicit constraints, while newer models are more robust to implicit ones.

The claims made in the submission are supported by extensive experiments.